# Chemical composition and *in vitro* nutritional assessment of watermelon (*Citrullus lanatus*) plant silage as a forage option for Murciano-Granadina goats

Zaira Pardo[1,2], Juan Manuel Palma-Hidalgo[1,2], Alberto Manuel Sánchez-García[1], A. Ignacio Martín-García[1]*

1 Estación Experimental del Zaidín, Consejo Superior de Investigaciones Científicas (CSIC), Granada, Spain, 2 Gut Microbiology Lab, Scotland's Rural College (SRUC), Edinburgh, United Kingdom

* ignacio.martin@eez.csic.es

## Abstract

Climate change is exerting significant negative impacts on various sectors, with livestock farming being particularly affected. One of the most pressing challenges in this context is the growing shortage in the availability of conventional fodder. This scarcity has intensified the search for alternative feed sources, with particular interest in underutilized resources often considered waste due to limited knowledge of their nutritional value. This study aimed to assess whether watermelon plant silage (WPS) could be used as a forage source in ruminants. The chemical composition of WPS and alfalfa hay (AH) was analyzed. Results showed similar protein content (21.1 *vs.* 18.9 g CP/100 g DM, respectively), with WPS exhibiting higher crude fat content (3.16 *vs.* 1.29 g/100 g DM) but lower hemicellulose (9.95 *vs.* 14.6 g/100 g DM) and cellulose (20.0 *vs.* 26.8 g/100 g DM) content compared to AH. In the first *in vitro* trial, WPS and AH were incubated independently to compare their fermentation behavior. WPS produced a higher concentration of short chain fatty acids (SCFA) (65.9 *vs.* 61.0 mM; $P = 0.304$), lower proportions of propionate ($P = 0.001$), and higher proportions of isobutyrate ($P = 0.001$). In a second *in vitro* trial, a formulated goat diet (commercial concentrate and AH in a 1:1 ratio) was used as a control to assess the impact of replacing 25% and 50% of AH with WPS. Trends towards higher value were observed in pH and $CH_4$ concentration as AH was replaced by WPS. The study concluded that WPS could serve as a viable fodder to replace AH in conventional goat diets, simultaneously reducing agricultural waste and serving as a regenerative model for implementing circular economy strategies in affected agronomic sectors.

**Data availability statement:** All the raw data are held in the Digital CSIC public repository in the site http://hdl.handle.net/10261/384534 (alternative links: https://digital.csic.es/handle/10261/384534; https://doi.org/10.20350/digitalCSIC/17197; https://digital.csic.es/bitstream/10261/384534/3/DatosPONE-D-24-50314%20DATA%203.pdf).

**Funding:** This study was supported by the Spanish Ministry of Science and Innovation through the project PID2020-119746RB-I00 (MCIN/ AEI/10.13039/501100011033), with A. Ignacio Martín-García serving as Principal Investigator. This work was carried out as part of the doctoral training of Alberto Manuel Sánchez-García, funded through a "Formación de Profesorado Universitario (FPU22/01066)" contract.

**Competing interests:** The authors have declared that no competing interests exist.

## Introduction

In recent years, the idea of the circular economy has gained importance to meet the requirements of sustainable livestock farming. In March 2020, the European Commission adopted a new Circular Economy Action Plan to increase the growth of sustainability that includes a "waste and recycling" policy aimed to protect the environment and human health [1]. The main targets of this policy are to stimulate innovation in recycling and improve waste management. These policies become increasingly more important because the world's population is expected to increase by 33% over the current population by 2050 [2], which will increase the demand for agricultural products [3], while the natural resources and provision of services from agriculture will not grow [4]. Furthermore, in response to the growing global population, there is a current proposal to shift animal feed towards products that are inedible for humans but suitable for livestock, such as crop by-products. This approach aims to replace human-edible products in animal diets, thereby increasing the overall food supply for human consumption [5]. In parallel, livestock is suffering from the negative effects of the warmer climate, which generates the scarcity of forage availability and, in consequence, the increase in prices, causing the need for searching alternative strategies to feed animals must be sought for the sector's economic sustainability [6]. Thus, including by-products in livestock feeding could help address the challenges of this emerging scenario by reducing the cost of feed and improving sustainability on-farm, both from the economic and environmental point of view [7].

Specifically, the biomass generated in the case of the watermelon plant was 100 million tons/year in 2019 worldwide [8]. Watermelon fruit and even its seeds have been assayed [9] but, to our knowledge the study of Hassan et al. [10], is the only one published on the suitability of using the watermelon plant as forage, and this has been carried out *in vitro* using sheep as an animal model for meat production systems. However, several studies concluded that there were substantial interspecific differences between goats and sheep in their ability to digest and utilize the nutrients of a number of by-products [11,12]. Another aspect to consider is that, the high moisture content may be the main challenge for the practical use and conservation of watermelon plants, because it usually contains certain amounts of discarded fruits left on the plant after harvesting. Nonetheless, ensiling is a technique usually chosen to address this problem and, in addition, it can also enhance the substrate's nutritional value and increase its acceptability [13]. Ensiling low-value agricultural residues represents an innovative strategy to generate high-quality feed for livestock, and its use can, at a minimum, maintain animal performance and health, while allowing the storage of large quantities of agricultural waste that provide a valuable source of feed for livestock when fresh forage is scarce [14]. We hypothesized that watermelon plant silage (WPS), because of their intrinsic properties, that resemble those of the medium-good quality forage [10], could be beneficial for dairy goats feeding. Alfalfa hay (AH) is the most widely available high-quality forage used in intensive dairy goat feeding. In the southern Iberian Peninsula, due to the region's climate, AH is typically dried as hay rather than ensiled, with hay being the preferred

form of feed for this livestock typology. However, AH faces an additional challenge: its availability and consequently its price are subject to significant variability and uncertainty, largely dependent on interannual climatic fluctuations and the impacts of global warming. This work aimed to study the nutritional value of WPS, as well as to assess the impact of replacing AH with WPS on different rumen *in vitro* fermentation parameters using goats as donors of inoculum of the rumen microbiota.

## Materials and methods

### Animals and feeding

Animal procedures and care were conducted in accordance with the Spanish regulations (RD 53/2013) that transpose the European Directive (2010/63/EU) on the protection of animals used in experiments or alternative scientific purposes. Experimental protocols were approved previously by the CSIC Ethical Committee for Experimental Animal Protection and authorized by the regional Andalusian government (procedure 06/07/2023/61) as the competent body in the matter. he experimental procedures did not involve causing pain or sacrificing animals. Four Murciano-Granadina goats (46 ± 4 kg body weight) fitted with rumen cannula were used in this study as donors of the inoculum to be used in the experimental procedures of *in vitro* ruminal fermentation. The handling of the animals did not involve causing them pain or sacrificing them, the procedure for extracting the rumen contents was of mild severity and their living spaces included environmental enrichment measures at all times. The animals were fed a diet consisting of a 50:50 ratio of concentrate to forage and had free access to water. This diet was administered in two equal meals at 08:00 and 14:00 h, providing sufficient quantities to meet their daily metabolic requirements [15]. The chemical composition of the concentrate (22% wheat shorts, 18% soybean meal, 17% barley, 15% maize, 10% beet pulp, 8% dehydrated alfalfa, 7% carob bean, 3% mineral-vitamin blend) and the forage (alfalfa hay) is shown in Table 1.

### Silage preparation

The watermelon growing area was fertilized with 175 kg/ha of an NPK mixture (ratio 2.6:0.8:3.4) before planting. The study utilized the entire watermelon plant (*Citrullus lanatus* Thunb. Sugar Baby variety) at its post-harvest phenological stage, encompassing branches and leaves but excluding roots. This plant material is typically considered agricultural waste and left in the field after fruit harvesting. In our case, the collected plant matter included small quantities of unharvested fruits that remained as surplus, deemed commercially insignificant. The WPS was prepared by pressing and wrapping with four to six layers of "bale wrap plastic" (25 µm stretch film). This was performed using a bale wrapper machine with a front-loader (Vicon RF 135 Balepack 3D Opticut 23, Brazil). Formic acid (0.45% of fresh matter) was previously added to facilitate the drop in pH. Bales were opened after 68 days of ensiling and pH was measured (3.61 ± 0.53). When opened, the colour, aroma and lack of mould were checked as an indicator of the quality of the ensiling process and some samples were squeezed to obtain separate juice fractions for the determination of the concentration of short-chain fatty acids (SCFA) as indicators of proper fermentation that ensure the absence of proliferation of clostridia or enterobacteria. Thus, the silage juice showed values of 2.86 ± 0.002 and 0.171 ± 0.005 g/100 g DM of WPS, respectively for acetate and propionate (Mean ± SD), while the values for the rest of the SCFA were negligible (< 0.04 g/100 g DM). Representative samples of watermelon plant and WPS were freeze-dried and ground before chemical analysis and *in vitro* experiments.

### Experimental design and samplings

Two *in vitro* trials were carried out using the methodology based on the non-renewed culture system of ruminal microorganisms [16] inoculated with rumen fluid from goats. In the first experiment, samples of WPS and AH were incubated independently to compare their fermentative behaviour. The AH used in both this experiment and the subsequent one was obtained from a mixture of samples collected from different locations on various bales of forage.

**Table 1. Nutrient composition of the materials used in the study.**

| Item | Watermelon plant | Watermelon plant silage | Alfalfa hay | Concentrate |
|---|---|---|---|---|
| DM[1], g/100 g FM[2] | 14.7 | 15.7 | 92.8 | 91.3 |
| Nutrients, g/100 g DM | | | | |
| OM[3] | 79.8 | 77.4 | 89.6 | 91.6 |
| CP[4] | 22.6 | 21.1 | 18.9 | 19.7 |
| CF[5] | 1.40 | 3.09 | 1.29 | 3.29 |
| NDF[6] | 36.7 | 36.3 | 49.4 | 30.3 |
| ADF[7] | 24.2 | 26.4 | 34.9 | 15.3 |
| ADL[8] | 7.86 | 6.40 | 8.01 | 4.75 |
| Hemicellulose | 12.6 | 9.91 | 14.6 | 15.0 |
| Cellulose | 16.3 | 20.0 | 26.9 | 10.6 |
| Total carbohydrates[9] | 55.8 | 53.2 | 69.4 | 68,0 |
| Non fibrous carbohydrates[10] | 19.1 | 16.9 | 20.0 | 38.3 |
| GGE[11], MJ/kg DM | 16.0 | 15.5 | 17.6 | 17.1 |
| Macrominerals, g/kg DM | | | | |
| Ca | | 44.7 | 4.40 | |
| K | | 22.5 | 11.1 | |
| Mg | | 6.07 | 2.30 | |
| P | | 2.91 | 2.00 | |
| S | | 2.28 | 1.30 | |
| Microminerals, mg/kg DM | | | | |
| Na | | 198 | 40.0 | |
| Fe | | 119 | 180 | |
| Al | | 103 | 227 | |
| Mn | | 39.6 | 43.1 | |
| Zn | | 14.6 | 46.8 | |
| Cu | | 5.62 | 5.33 | |
| Ti | | 3.42 | 10.6 | |
| As | | n.d.[12] | 0.340 | |
| B | | n.d.[12] | 11.9 | |
| Si | | n.d.[12] | 129 | |
| Sr | | n.d.[12] | 46.0 | |

[1]DM: Dry matter;

[2]FM: Fresh matter;

[3]OM: Organic matter;

[4]CP: Crude protein;

[5]CF: Crude fat;

[6]NDF: Neutral detergent fiber;

[7]ADF: Acid detergent fiber;

[8]ADL: Acid detergent lignin;

[9]Total carbohydrates = (100-(CP + CF + Ash);

[10]Non fibrous carbohydrates = Total carbohydrates – NDF;

[11]GE: Gross energy;

[12]n.d.: not detected.

A complete randomized design was used in the first trial, and the main effect was the type of forage (AH and WPS). The same amount of DM (300 mg) of each forage was carefully weighed into 120 ml Wheaton bottles. Twelve bottles were prepared per type of forage (four sources of inoculum in triplicate). Approximately 500 ml of rumen content was collected from the rumen of four fistulated fasting (12 h) goats and filtered through four layers of cheesecloth. The rumen fluid of each goat (n = 4) was mixed separately in 1:3 proportion with buffer solution [17]. One bottle per buffered inoculum was incubated as blank (without diet). Then, 40 ml of culture media in bottles containing either AH or WPS were added and incubated in triplicate for each donor animal. The bottles were incubated at 39 ˚C for 72 h and the gas pressure in each bottle was measured using a wide range pressure meter (Sper Scientific LTD, Scottdale, AZ, USA) at 2, 4, 6, 8, 12, 24, 48 and 72 h after the start of incubation. These data were used to calculate the kinetics of gas production from forage fermentation as described later. The gas production (GP, ml) was measured using a system consisting of a 40 ml capacity syringe coupled to the pressure meter via a three-way stopcock. This apparatus was connected to a needle inserted into the bottle's headspace. The gas was extracted using the syringe until the system's pressure equalized with atmospheric pressure and then the volume (ml) was noted. After 24 h of incubation, one of the three initially prepared set of bottles was opened, the pH was measured, using a pH-meter (Crison GLP 22, Hach Lange, Barcelona, Spain), and samples (0.80 ml) of each bottle were obtained for SCFA determination following the methodology described by Arco-Pérez et al. [7]

The second *in vitro* trial evaluated the effect of replacing AH with WPS in different proportions using a completely randomized design, and the main effect was the substitution rate: 0, 25 and 50%. The samples weighted into the Wheaton bottles were: 1) Control diet consisting in 300 mg of dairy goats-formulated diet of AH and commercial concentrate (50:50 ratio), 2) the same diet as before diet but where 25% of the AH was replaced by WPS, 3) a diet where 50% of the AH was replaced by WPS. One bottle per buffered inoculum was incubated as blank (without diet). *In vitro* incubations were prepared following the same protocol as in the first trial, with 9 bottles per level of AH substitution by WPS (three inoculum sources in triplicate). Each bottle contained 300 mg DM of the respective diet and 40 ml of incubation solution. The bottles were incubated at 39 °C for 72 hours. Gas pressure measurements were taken at the same time intervals and following the same procedure as in the initial experiment. Additionally, after 24 h of incubation, one bottle from each triplicate set was opened to measure pH, and samples were collected for SCFA quantification.

## Chemical analyses

The dry matter (DM) (method 934.01), organic matter (OM) (method 942.05) and crude fat (CF) (method 920.39) in both WPS and AH were evaluated in triplicate according to the AOAC procedures [18]. The nitrogen (N, AOAC method 990.03) was determined by Dumas procedure (Leco TruSpec CN®, St. Joseph, Michigan, USA) to obtain the crude protein (CP = total N g/100 g DM × 6.25). Neutral (NDF) and acid (ADF) detergent fibre were analyzed following the sequential procedure of Van Soest [19] using the Ankom 220 Fiber Analyzer (ANKOM Technology). The cellulose was solubilized with 72% sulfuric acid for acid detergent lignin (ADL) determination. These fibre fractions were expressed excluding residual ash. The total carbohydrates amount was calculated taking into account the proportion of ashes, CP and CF, using the following formula: Total carbohydrates (%) = DM – (Ashes + CP + CF). The non fibrous carbohydrates were calculated using the following formula: Non fibrous carbohydrates (%) = Total carbohydrates – NDF. Finally, the gross energy (GE) content was measured determined by using an adiabatic calorimeter (Parr Instruments Co. model 1356, Moline, IL, USA).

Heavy metals and mineral contents were determined by atomic spectroscopy inductively coupled plasma-optical emission spectrometry (ICP-OES) analysis. The analyses were carried out in duplicate. The content of some essential and non-essential minerals was estimated. The method was based on the addition of a mixture of Milli-Q water and 65% $HNO_3$ to the sample (1:3), followed by digestion in an ultra-wave digester at 220 ˚C for 15 min and cooling to 60 ˚C under high pressure. The product was then transferred to a volumetric flask and diluted with Milli-Q water. Determinations were conducted using an ICP-OES 720-ES system with a sea spray nebulizer and axial torch. Two spectral lines per element

                                    

were selected, with linearity (correlation coefficient ≥0.995) ensured by utilizing standard solutions. Each analytical result was obtained by calculating the average of the two spectral line readings.

The amino acid (AA) composition of forages was determined by high-performance liquid chromatography using a Waters Lambda-Max LC Spectrophotometer detector (Waters Corporation, USA) applying the Waters® Pico-Tag method, which involves precolumn derivatisation with phenylisothiocyanate. Protein hydrolysis was carried out with 6 N HCl using evacuated tubes at 110 ˚C for 24 h [20].

Rumen fluid samples were analysed for total SCFA concentration as well as their individual molar proportions. Immediately after opening the bottles at 24 h of incubation, a 0.8 ml aliquot of incubation fluid was mixed with an equal amount of an acid solution consisting of metaphosphoric acid (20% wt/vol in 0.5 N HCl) and crotonic acid (0.8 g/L, internal standard), then the samples were centrifuged at 2700 g for 20 min. The SCFA concentration was determined using a gas chromatography (GC) system coupled with a Flame Ionization Detector (Autosystem Perkin-Elmer Cor., Norwalk, Connecticut, USA).

To measure the methane ($CH_4$) production, 4 ml of headspace gas were taken at 24 h at atmospheric pressure and stored in an evacuated tube (Terumo Europe N.V., Leuven, Belgium) at 4 °C until the determination by GC using a HP Hewlett 5890 Packard Series II gas chromatograph (Waldbronn, Germany) equipped with a flame ionization detector and with the methodology described by Kheddouma et al. [21].

### Statistical analysis and calculations

Data analysis was performed using the SPSS software (IBM Corp. IBM SPSS Statistics for Windows, Version 29.0.0.0 Armonk, New York USA). The comparison between AH and WPS fermentation parameters was analysed by a one-way analysis of variance (ANOVA). Normality and homoscedasticity assumptions for ANOVA were verified using the Shapiro-Wilk test ($P > 0.10$ for all variables) and Levene's test ($P > 0.05$), respectively. The effect of substituting AH with WPS was also analysed with ANOVA, followed by orthogonal polynomial contrast to evaluate linear trends in the effect of increasing substitution levels. These analyses examined the impact of forage type and substitution levels on degradation parameters (GP, ME, OMD, pH, SCFA, and $CH_4$ production). Fisher's Least Significant Difference (LSD) test was selected for planned pairwise comparisons following significant ANOVA results ($P < 0.05$). No unplanned comparisons were conducted, maintaining the per-comparison error rate ($α = 0.05$). Trends were noted for $P$ values between 0.05 and 0.10.

The gas dynamics generated throughout the incubation was adjusted using the exponential model $y = A·(1-e^{-ct})$ described by France et al. [22], where "$y$" represents the gas production (GP, ml/g); "$A$" represents the asymptote (ml); "$c$" represents the rate of gas production ($h^{-1}$) and "$t$" represents the time of incubation (h). Feed chemical composition (CP and OM) and fermentation GP data were used for the estimation of the metabolizable energy (ME) and organic matter digestibility (OMD) following the models described by Menke and Steingass [17]:

$$ME\ (MJ/kg\ DM)\ =\ 2.20\ +\ 0.136GP\ +\ 0.0057CP\ +\ 0.00029\ CP^2$$

$$OMD\ (g/kg\ OM)\ =\ 148.8\ +\ 8.89GP\ +\ 0.448CP\ +\ 0.651Ash$$

The $CH_4$ production rate relative to SCFA production (ml $CH_4$/mmol SCFA) was calculated as the ratio between $CH_4$ and SCFA production, both expressed per gram of incubated DM.

## Results

### Chemical and amino acid composition

The chemical and AA composition of WPS and AH are shown in Tables 1 and 2. The DM content was low, both in WPS and in the starting material used for the ensiling process (15.7 and 14.7 g/100 g fresh matter (FM), respectively),

compared to the value obtained for the properly hayed alfalfa (92.8 g/100 g FM) used in this comparative study. The OM varied from 77.4 (WPS) to 89.6 g/100 g DM (AH). The CP was also variable, with values from 21.1 (WPS) to 18.9 g/100 g DM (AH). In the same way, CF varied widely, from 3.09 (WPS) to 1.29 g/100 g DM (AH). The content in NDF varied, from 36.3 (WPS) to 49.4 g/100 g DM (AH), ADF from 26.4 (WPS) to 34.9 g/100 g DM (AH) and ADL from 6.40 (WPS) to 8.01 g/100 g DM (AH). Consequently, hemicellulose ranged from 9.91 (WPS) to 14.6 g/100 g DM (AH) and cellulose from 20 (WPS) to 26.9 g/100 g DM (AH). Total carbohydrates changed from 53.2 (WPS) to 69.4 g/100 g DM (AH) and Ngross energy (GE) from 15.5 (WPS) to 17.6 MJ/kg DM (AH). A large amount of variation between diets was observed in the macrominerals content, with WPS having a higher Ca content (44.7 g/kg) compared to AH (4.40 g/kg). Further-more, the Mg and P content was higher for WPS (6.07 and 2.91 g/kg, respectively) compared to AH (2.30 and 2.00 g/kg, respectively).

Regarding the AA composition, the content in relation to the DM was lower for WPS compared to AH (151 and 165 g AA/kg DM, respectively) and up to 18% lower with respect to the total N content (716 and 874 g AA/kg N, respectively for WPS and AH). The WPS had a higher content of glutamic acid, phenylalanine, serine, threonine and leucine, while the content in glycine + histidine, aspartic acid, proline, lysine, tyrosine, arginine, isoleucine and valine were higher for AH. However, the sum of essential (EAA) and non-essential (NEAA) amino acids was balanced and similar for both forages (48.8 and 50.3 g EAA/100 g AA and 51.2 and 49.7 g NEAA/100 g AA, respectively for WPS and AH) Table 2.

**Comparison between both forages based on *in vitro* ruminal fermentation parameters**

The results of the fermentation parameters obtained from the first *in vitro* trial with AH and WPS incubated independently are shown in Table 3. In the present study, we found no significant differences. ($P = 0.874$) between the forages in the

Table 2.  Amino acid (AA) composition of forage protein compared.

| | Watermelon plant silage | Alfalfa hay |
|---|---|---|
| g AA/100 g AA | | |
| Aspartic acid | 8.56 | 12.5 |
| Serine | 5.86 | 4.69 |
| Glutamic acid | 14.9 | 9.66 |
| Glycine+Histidine* | 4.96 | 8.15 |
| Arginine | 5.67 | 6.15 |
| Threonine* | 5.38 | 4.63 |
| Alanine | 8.10 | 6.20 |
| Proline | 8.10 | 10.5 |
| Tyrosine* | 3.22 | 3.92 |
| Valine* | 6.61 | 6.67 |
| Lysine* | 5.88 | 7.40 |
| Isoleucine* | 4.59 | 4.84 |
| Leucine* | 9.16 | 8.66 |
| Phenylalanine* | 9.03 | 5.98 |
| EAA[1] | 48.8 | 50.3 |
| NEAA[2] | 51.2 | 49.7 |
| g AA/kg DM | 151 | 165 |
| g AA/kg N | 716 | 874 |

[1]EAA: essential amino acids [23]; Cys, Met and Trp were not determined);

[2]NEAA: non-essential amino acids.

GP andthe asymptotic value of total gas production (A, ml) ($P$ = 0.687). Nevertheless, differences were found in the rate of degradability (c, $h^{-1}$), which was lower for the AH compared to WPS ($P$ = 0.014) and a trend for the pH of the medium after 24 h of incubation, being lower for AH compared to WPS (6.96 and 7.01, respectively; $P$ = 0.063). Furthermore, the metabolisable energy (ME) did not showed significant differences between the AH and WPS (7.72 and 7.37 MJ/kg DM, respectively; $P$ = 0.157). Hence, OMD was similar for AH and WPS (511 and 494 g/kg, respectively; $P$ = 0.300).

There was a trend in the $CH_4$ production to be lower for the AH compare to the WPS (87.4 and 94.8 ml/l GP, respectively; $P$ = 0.0.072). According to the total SCFA concentration, the fermentability of both forages were similar (61.0 and 65.9 mM, for AH and WPS respectively; $P$ = 0.304). In terms of SCFA molar proportions, no differences were detected in the acetate, whereas propionate was higher for AH than WPS (19.7 and 17.6%, respectively; $P$ = 0.001). Conversely, isobutyrate was lower for AH compared to WPS (1.22 and 1.70%, respectively; $P$ = 0.001). The isovalerate proportion was lower with the AH incubation compared to WPS (1.59 and 1.97%, respectively; $P$ = 0.011). No significant differences were found in the molar proportions of butyrate and valerate ($P$ = 0.457). Consequently, the acetate:propionate ratio was lower in AH fermentation compared to WPS (3.61 and 4.08%, respectively; $P$ = 0.010).

The graph of the accumulated GP for both WPS and AH forages is represented in Fig 1. The WPS had the highest cumulative GP compared to AH, regarding the asymptote of the kinetics curve. Initially, the GP slope was similar for both WPS and AH forages, but after 20 hours of fermentation, the WPS GP extent exceeded the threshold of the AH one.

**Table 3. Fermentation parameters, after 24 h of *in vitro* culture of rumen microorganisms, of the compared forages.**

| | Alfalfa hay | Watermelon plant silage | SEM[1] | *P* value |
|---|---|---|---|---|
| A[2], ml | 132 | 128 | 4.85 | 0.687 |
| c[3], $h^{-1}$ | 0.101 | 0.117 | 0.003 | 0.014 |
| pH | 6.96 | 7.01 | 0.013 | 0.063 |
| GP[4] 24h, ml/g DM | 119 | 118 | 4.34 | 0.874 |
| ME[5], MJ/kg DM | 7.72 | 7.37 | 0.122 | 0.157 |
| OMD[6], g/kg | 511 | 494 | 7.95 | 0.300 |
| $CH_4$, ml/l GP | 87.4 | 94.8 | 1.97 | 0.072 |
| Total SCFA[7], mM | 61.0 | 65.9 | 2.34 | 0.304 |
| Acetate, % | 69.2 | 70.2 | 0.451 | 0.308 |
| Propionate, % | 19.7 | 17.6 | 0.256 | 0.001 |
| Butyrate, % | 6.91 | 7.23 | 0.218 | 0.477 |
| Isobutyrate, % | 1.22 | 1.70 | 0.040 | 0.001 |
| Valerate, % | 1.38 | 1.33 | 0.030 | 0.437 |
| Isovalerate, % | 1.59 | 1.97 | 0.067 | 0.011 |
| Acetate/Propionate | 3.61 | 4.08 | 0.081 | 0.010 |

[1]SEM: Standard Error of Mean;

[2]A: Asymptotic value of total gas production;

[3]c: the rate of degradability;

[4]GP: gas production;

[5]ME: Metabolizable energy;

[6]OMD: Organic matter digestibility;

[7]SCFA: Short-chain fatty acids.

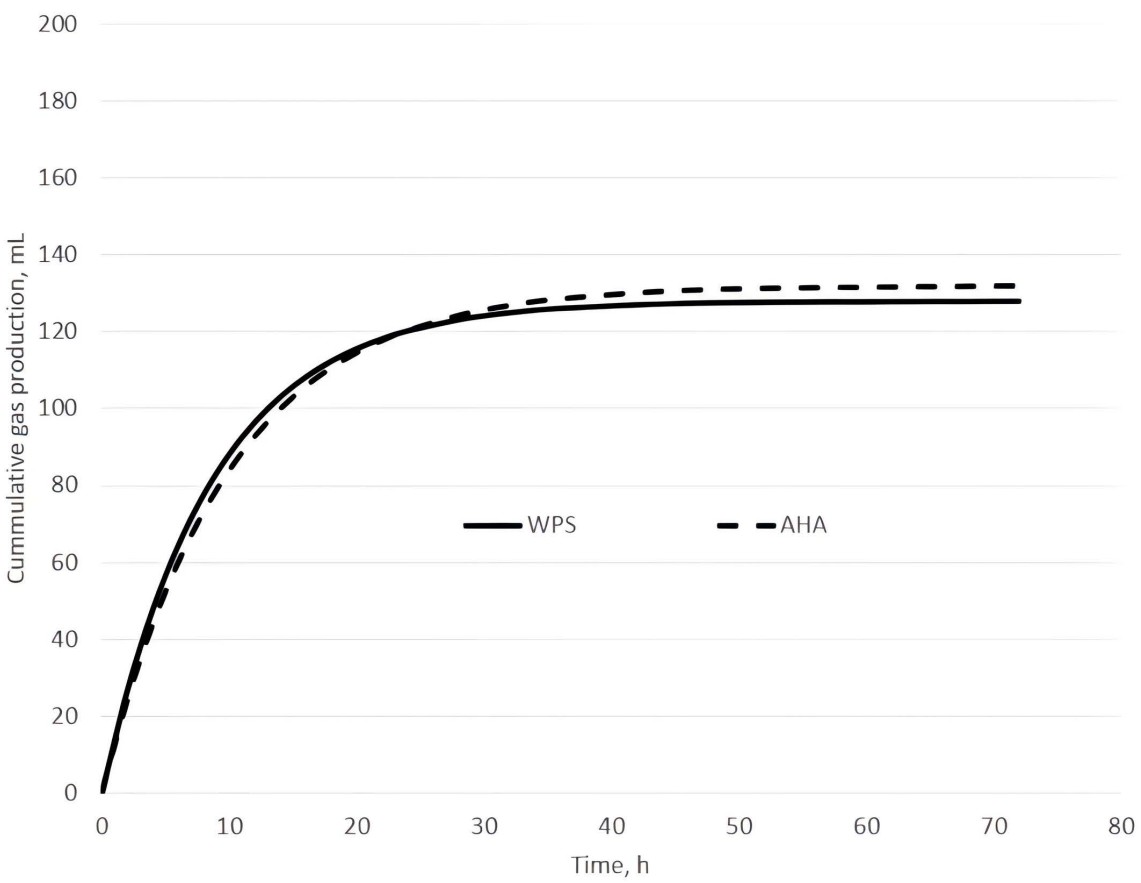

**Fig 1. Cumulative gas production in the *in vitro* incubation at 72h for WPS and AH.**

### Effects of replacing alfalfa hay with watermelon plant silage on the ruminal fermentation parameters of the experimental diet

The outcomes of the second experiment, which assessed the effects of substituting AH with WPS, are presented in Table 4. The replacement of 25 or 50% of AH with WPS did not induce significant differences ($P > 0.05$) in terms of GP at 24 h, the potential or asymptote of GP (parameter A) over time, the ME of the diets, the OMD of the diets, the total SCFA concentration and the molar proportions of propionate, butyrate, valerate and the ratio of acetate:propionate. Nevertheless, significant differences in the molar proportion of acetate, isobutyrate and isovalerate were found when AH was replaced with WPS (25 and 50%). A decrease in the proportion of acetate and isobutyrate was observed with the inclusion of WPS. Nonetheless, the proportion of isovalerate increased when the percentage of WPS increased. No clear effect of the substitution on the $CH_4$ production rate as a function of SCFA production was observed either. Regarding the $CH_4$ concentration, a tendency to increase was observed as the percentage of WPS increased ($P = 0.053$).

The effect of replacing different percentages of AH with WPS in GP is graphically represented in Fig 2. Diets with 25 and 50% substitution of AH with WPS showed a slightly higher GP asymptote (A), while the rate of GP until 20 h of incubation resulted quite similar among treatments.

**Table 4. Effect of substituting different percentages of alfalfa hay (0, 25 and 50%) with watermelon plant silage on fermentation parameters after 24 h *in vitro* rumen microorganisms culture.**

| | Substitution rate, % | | | SEM[1] | *P* value |
|---|---|---|---|---|---|
| | 0 | 25 | 50 | | |
| A[2], ml | 176 | 177 | 177 | 2.90 | 0.984 |
| c[3], h$^{-1}$ | 0.134 | 0.133 | 0.138 | 0.003 | 0.684 |
| pH | 6.96 | 6.94 | 6.90 | 0.009 | 0.011 |
| GP[4] 24h, ml/g DM | 164 | 167 | 167 | 2.69 | 0.922 |
| ME[5], MJ/kg DM | 8.84 | 8.86 | 8.83 | 0.060 | 0.984 |
| OMD[6], g/kg | 583 | 585 | 584 | 3.95 | 0.978 |
| $CH_4$, ml/l GP | 100 | 110 | 110 | 1.76 | 0.053 |
| $CH_4$, ml/mmol SCFA | 66.9 | 81.2 | 70.6 | 3.82 | 0.311 |
| Total SCFA[7], mM | 66.3 | 67.2 | 66.7 | 0.246 | 0.387 |
| Acetate, % | 21.6 | 20.6 | 20.7 | 0.153 | 0.047 |
| Propionate, % | 7.40 | 7.55 | 7.79 | 0.224 | 0.815 |
| Butyrate, % | 1.36 | 1.35 | 1.43 | 0.020 | 0.129 |
| Isobutyrate, % | 1.56 | 1.53 | 1.46 | 0.017 | 0.047 |
| Valerate, % | 1.80 | 1.75 | 1.86 | 0.034 | 0.316 |
| Isovalerate, % | 3.05 | 3.26 | 3.22 | 0.029 | 0.024 |
| Acetate/Propionate | 176 | 177 | 177 | 2.90 | 0.984 |

[1]SEM: Standard Error of Mean;

[2]A: Asymptotic value of total gas production;

[3]c: the rate of degradability;

[4]GP: gas production;

[5]ME: Metabolizable energy;

[6]OMD: Organic matter digestibility;

[7]SCFA: Short-chain fatty acids.

## Discussion

The use of plant silages as an alternative source in animal feed has been considered during the last decades, showing a wide variation in its chemical composition due to several factors such as plant species, climatic conditions during cultivation and treatments to improve ensiling processing. The principle of ensiling is based on preserving green fodder under anaerobic conditions to support the growth of lactic acid-producing bacteria, which generate lactic acid and cause a decrease in the pH of the conserved material [24].

Several studies have shown promising results when using plant silages to feed ruminants [25,26], but is scarce the scientific literature on the suitability and adequacy of the watermelon plant ensiling, especially considering the entire plant and not only surplus fruits. According to the literature [27], maintaining upper limits of 3% acetate and 0.5% propionate in the squeezed silage liquid has been established as a good indicator of adequate fermentation of the silage and the absence of clostridia or enterobacteria proliferation. In the WPS, the percentages of acetate and propionate were 2.86% and 0.171%, respectively. These authors also argued that high concentrations of these acids are related to a low voluntary feed intake of silage in cows, although it may not be directly attributable to those compounds, rather because they also can be considered indicators of an incorrect conservation or storage. In any case, SCFA concentrations in silage are proportional to the moisture content, that in the case of WPS (almost 85%, conferred by the presence of watermelon fruits in the pre-silage mixture) was much higher than the average of the studies reviewed in the cited work (75% upper limit).

 

Similar to what Lin et al. [28] found when analyzing the impact of AH silage on the its chemical composition, the nutrient composition of the WPS hardly changed concerning what the plant had before the ensiling process. They reported a significant reduction in water-soluble carbohydrates, probably for a reason that can explain the decrease in hemicellulose in exchange for the increase in cellulose (in a magnitude close to 20%) observed in our study. The main explanation could be that hemicellulose could be susceptible to being used to a greater extent by certain microorganisms involved in the fermentation that occurred during the ensiling process.

Some authors reported variations in the saturation of fatty acids due to the ensiling, but there is no available information on the impact of this process on the plant CF content that could explain the increase (by more than 100%) of this nutrient observed in the WPS. However, Hassan et al. [10], reported a WPS CF content similar to that in our product (2.80 *vs.* 3.09 g/100 g DM, respectively), whereas Ibrahim et al. [29] found a range of CF content in the intact watermelon plant (from 1.19 to 1.82 g/100 g DM) which covered the value of our observation (1.40 g/100 g DM) before ensiling.

Alfalfa stands out among forage crops due to its high levels of CP and energy, which minimize the necessity for additional supplements in feed and makes it ideal for inclusion in the diets of high-yield dairy ruminants. The comparison between the composition of WPS and AH used in our trial reflected that the former showed a higher content of minerals, CP and CF, while AH contained higher NDF, ADF and ADL proportions. Considering this, and that the NDF content (nearly 50%) of the AH used in our study, it can be regarded as a medium-quality forage legume, while WPS could be expected to offer a higher potential for providing ME. The ADF represents the percentage of highly indigestible and slowly digestible components in forages, including cellulose, lignin, pectin, and ash. A lower ADF value in WPS, approximately 25% less than in AH, suggests that this forage might be more digestible and thus suitable for dairy goats.

The lower OM and carbohydrates content in WPS likely resulted from the presence of earth in the obtained plant samples, suggesting that the collection method used in this experiment may have been suboptimal. This same reason would

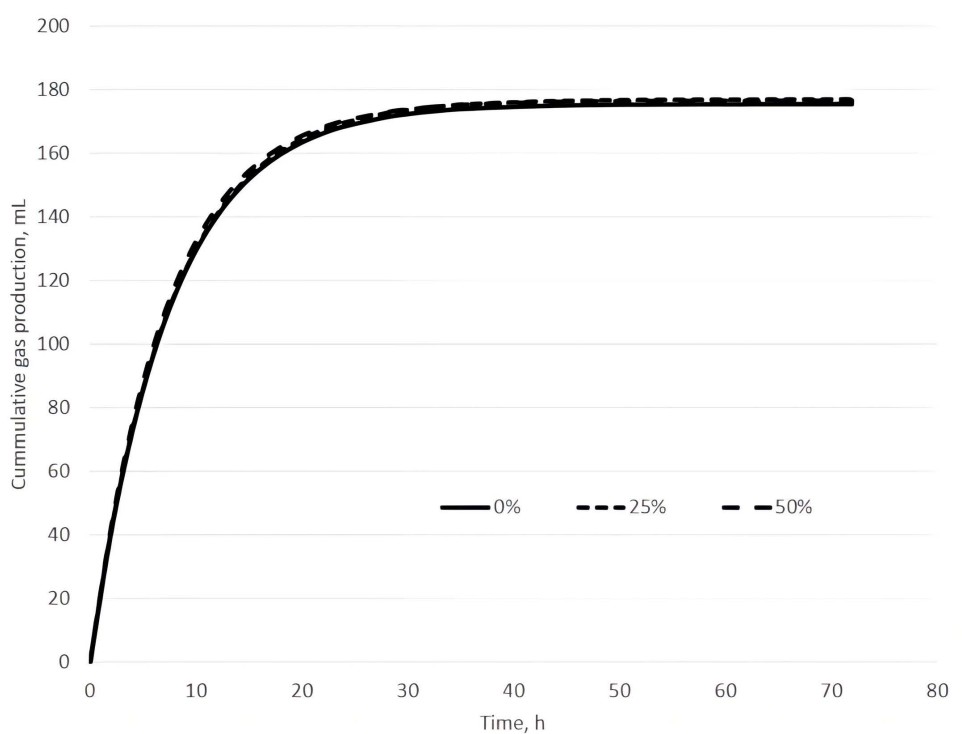

**Fig 2. Cumulative gas production when different percentages of AH were replaced by WPS at 72 h.**

explain the very high calcium (Ca) content since the WPS cultivation soil in this study was made up of quaternary deposits of the Ca luvisol type. Minerals are necessary for metabolic processes in small ruminants and Ca is required for lactation, maintenance and growth in small ruminants. Those requirements have been reported to be between 1.2 to 2.6 g/kg DM [30] and, while AH is considered rich in Ca source (4.40 g Ca/kg DM), the concentration of this element in WPS exceeded almost twenty times the upper threshold of the requirements range (44.7 g Ca/kg DM in WPS). However, if the Ca is in the form of carbonates of soil contamination, the bioavailability of this element may be lower and, adversely, it could depress the absorption of Fe [31]. Nevertheless, Ca is one of the primary nutrients in ruminant diets, following protein and energy in importance. It plays crucial roles in maintaining acid-base balance, facilitating muscle contraction, transmitting nerve impulses, and contributing to protein biosynthesis, especially in lactating animals [32]. Calcium is also essential for skeletal mineralization, and its adequate supply can prevent diseases such as osteomalacia, osteoporosis, and fibrous osteodystrophy [33]. Most plant-based foods, except for leguminous forages, have low Ca content. Consequently, proper Ca supplementation in ruminant diets typically relies on animal-derived and inorganic sources, which have a bioavailability greater than 66% [34]. The high Ca concentration in WPS could potentially serve as an effective alternative source. Thus, in a relevant way WPS could reduce the costs associated with dietary Ca supplementation in ruminant nutrition.

The WPS showed higher concentrations of K, Mg and Na, and lower of Fe, Al, Zn, Ti, As, B, Si and Sr than AH. In comparison with AH, and contrarily to described by Hassan et al. [10], the low heavy metals content of WPS draws attention, which would constitute it as a safe fodder for animals.

The AA analysis plays a crucial role in examining the make up of proteins, as well as in studying the constituents of foods and animal feeds and their potential to cover the protein requirements of the different animal species. Moreover, EAA is defined as AA whose carbon skeletons are not synthesized *de novo* by animal cells or AA that are insufficiently synthesized *de novo* by animal cells relative to metabolic needs [23]. Although many species of rumen bacteria are capable of *de novo* synthesizing AA, ruminants' diets must provide sufficient protein and EAA when high rates of growth or lactation are required [35]. Ibrahim et al. [36] studied the AA composition of different parts of the watermelon plant (*Citrullus vulgaris*) and their results regarding proportions of AA in stem and leaves can be considered equivalent to those found in our study, including the observation related to a proportion higher in NEAA than in EAA. They also concluded that most of the AA values of the watermelon plant are comparable with those of most vegetable proteins, an observation that can also be extended to the comparison of WPS with AH performed in our study. Nevertheless, it should be noted that the ratio of AA-N in WPS was 18% lower than in AH, probably indicating a worse protein value for WPS. This limitation could potentially be offset by the higher total N content (exceeding 11.5%) of this alternative fodder, which might be suitable for promoting microbial synthesis in the rumen.

The *in vitro* gas technique, which relies on volume and pressure measurements, has been used for decades to assess the degree of fermentation of diets in ruminants [37] and monogastric animals [38]. It serves as an initial approach for the evaluating the nutritional potential of novel and alternative sources of feed for animal production. Using the *in vitro* gas technique allows to reduce the cost, time and use of *in vivo* experiments [39] and is considered a replacement and reduction system to implement two of the three Rs (replacement, reduction and refinement) principles to address the ethical protection of experimental animals. For this reason, this technique is currently widely used in animal nutrition laboratories, and its methodological application continues to evolve and improve [39].

When evaluating the nutritive value of unexplored feeds, it is important to consider the rumen fermentation parameters, which adequacy indicates that it can be conveniently degraded by the animal and supplies the required nutrients. In the present study, in terms of rumen fermentation dynamics, the WPS was shown to lead a fermentative process similar to AH regarding the asymptotic limit (A) of GP, indicating the suitability of its inclusion in the diet of goats. Although the range of ME values for medium quality AH reported in the literature, determined through *in vivo* tests, typically falls between 8 and 9 MJ/kg DM, and differs from the 7.72 MJ/kg DM found in this study, it is important to recognize that the *in vitro* values obtained are still valid for comparing the energy availability of AH and WPS. This allows for an assessment of the

suitability of WPS as a potential substitute for AH. Nonetheless, the ME calculations following the model proposed by Menke and Steingass [17], which considered the GP and the chemical composition (CP and OM) of the substrates, did not revealed significant differences for the AH in comparison to WPS. Thus, the underestimation obtained in the calculation of ME was due to the low GP records, which resulted from a lower ruminal inoculum to buffer ratio in our study (1:3) compared to that used in the work of mentioned authors (1:2).

The end products of dietary carbohydrate fermentation in ruminants are SCFA, mainly acetate, propionate and butyrate [40]. All of them are the main source of energy in ruminants, accounting for more than 70% of total ME [40]. Acetate and propionate are essential for fat synthesis and gluconeogenesis [41]. In the present study, when comparing WPS to AH *in vitro* fermentation, we found that WPS fermentation promoted similar acetate and butyrate proportions to the AH but lower propionate rate. Considering the total SCFA production, the energy potential showed by the rumen fermentation of the WPS was equivalent to AH. Total SCFA production is typically expressed in relation to digested OM as an indicator of fermentation efficiency, and such calculation turned out to be completely equivalent for both types of forage, with values of 24.2 and 23.8 mmol of SCFA/g of digested OM for WPS and AH, respectively. When examining the $CH_4$ proportion in the total GP, a trend to increase the methanogenic potential of WPS compared to AH. The isobutyrate and isovalerate proportions were lower for AH compared to WPS. These SCFAs are iso-acids and are involved in the stimulation of microbial protein synthesis [42].

The second trial did not reflect a greater impact at the levels of 25 and 50% replacement of AH by WPS on rumen fermentation parameters, except for trends to linearly increase the methanogenic activity of the diet. The $CH_4$ production in the rumen is mediated by the release $CO_2$ and hydrogen, which are generated from the synthesis of SCFA during carbohydrate fermentation, and represents an energy loss for the animal. An efficiency index for this process involves relating methanogenesis to the SCFA production derived from diet fermentation. Thus, regarding the observed trend towards increased methanogenicity of the diet with the inclusion WPS, it is important to note that no significant differences were observed in the aforementioned index (ml $CH_4$/mmol SCFA). This suggests that the trend may be attributed to shifts in the fermentation process extent rather than to greater fermentation inefficiency.Based on the results obtained in Experiment 1, an increase in both ME and total SCFA production might have been awaited during Experiment 2 due to the inclusion of WPS in the diet. However, this effect was not observed, likely due to synergistic interactions that occurred in the utilization of the nutrient combination from forages and concentrate by ruminal microorganisms.

Regarding the decrease in acetate as we increased the inclusion of WPS, similar results have been found in an *in vivo* study in cows where mulberrry was included up to 10%, which had a similar composition to our WPS in terms of ADF and NDF, reporting a decrease in acetate formation [43].

Isovalerate is a SCFA produced in ruminal fermentation from leucine, which has been found in high proportions in WPS. This fact could be responsible for an increase in isovalerate in the experiment. However, these differences were not observed when comparing the *in vitro* ruminal fermentation of WPS to AH, so the mechanism by which this variation in these isoacids occurs is not entirely clear.

In an *in vivo* trial, Soliman et al. [44] used a mixed ration for lactating cows (40:60 forage to concentrate ratio) to study the replacement of berseem (leguminous plant) hay by dried watermelon vine in different percentages (0, 25, 50, 75 and 100%) and observed, contrary to our observations, that OMD and total SCFA production decreased from a 50% of hay replacement. From this substitution level, they also observed a detrimental effect on productive parameters (i.e., milk yield and its dry extract). Nevertheless, it is necessary to consider that the quality of the watermelon plant used by these authors had significantly lower nutritional characteristics compared to the WPS in our study, as it had lower protein contents and higher cellulose and, especially, hemicellulose contents.

Recently, Hassan et al. [10], studied the impact of the replacement (10, 20 and 30%) of sunflower meal (protein concentrate that represented 6% of the diet) with WPS using an *in vitro* approach with rumen inoculum from sheep and no effect was observed on GP or total and molar proportions of SCFA. This agrees with our results, although the total

proportion of inclusion of WPS in the diet was much higher in our case, which indicates that a considerable replacement of the standard diet with WPS has no harmful effects on rumen fermentation.

As global dairy consumption continues to grow, sustainable livestock production becomes increasingly crucial. Besides, climate change and geopolitical instability are significantly impacting forage production and availability, leading to reduced yields, lower quality, and higher prices. These factors are threatening the economic sustainability of livestock farms. In this context, alternative feed sources like WPS could offer substantial cost savings, as they require minimal production costs beyond collection and transport [45,46]. The magnitude of this reduction is indeed variable in the global dairy goat sector, and forage typically accounts for 25–50% of the total feed ration which costs is particularly significant in dairy goat farming. To provide a representative reference, in our study area (Andalusia), Morales-Jerrett et al. [47] found that feed costs (including concentrates and fodder) represent 58.5 ± 3.9% of the total cost structure for dairy goat farms of different typologies.

Dairy goat production is an efficient system to transform the energy contained in waste feedstocks to good quality foods through the contribution of microbial protein synthesis [48,49]. Utilizing WPS offers significant environmental benefits by repurposing food industry waste to reduce landfill waste and promote a circular economy. As with other by-products, compared to conventional forage, WPS requires fewer resources, potentially reducing carbon footprint and greenhouse gas emissions [45]. This approach addresses waste management challenges while contributing to more sustainable livestock farming practices in the face of climate change and resource constraints.

Based on the results of this initial *in vitro* nutritional evaluation of WPS, which suggest its potential as an alternative forage, further *in vivo* evaluations are needed due to the inherent limitations of this approach, particularly regarding intake, absorption and animal performance or production. First, animal trials should be designed to study *in situ* ruminal degradability, particularly of important WPS nutrients such as protein, as well as the intestinal availability of rumen-undegraded protein. Furthermore, it is crucial to determine the energy and nutrient balance in animals where WPS constitutes a substantial portion of the forage fraction in their diet, at levels comparable to the highest proportions tested in this study. Finally, an on-farm evaluation of WPS inclusion in dairy goat diets would be highly beneficial to assess its effects on animal performance, milk production and composition, as well as its impact on the farmer's economy and the carbon footprint of the final product.

## Conclusions

This study shows that the nutritional value of watermelon plant silage suggests it may be suitable for use as a forage component in the formulation of diets for dairy goats. This claim would be based both on its chemical composition and on the fact that, compared to alfalfa hay as a reference forage, its inclusion in the ration did not alter the different parameters of *in vitro* ruminal fermentation and the supply of nutrients and energy. This evidence supports the use of watermelon plant silage as an alternative source in ruminant feeding, although *in vivo* studies are required to elucidate the influence of this fodder on voluntary feed intake and animal performance and production.

In addition, the inclusion of watermelon plant silage would contribute to the circular bioeconomy by utilizing agricultural waste as a feed source for livestock. This approach would help alleviate the problem of shortage of conventional fodder while potentially reducing costs for livestock farmers.

## Acknowledgments

The authors thank the Technical Scientific Services "Animal Experimentation" and "C and N Analysis" of the Estación Experimental del Zaidín (CSIC). Also, thanks for their help in laboratory procedures and advice to I. Jiménez and E. Jiménez. We extend our gratitude to F. Martínez, Veterinary Technician at Los Filabres Coop, for his initial request highlighting the need for this study and for providing us with the opportunity to obtain the raw materials.

## Author contributions

**Conceptualization:** A. Ignacio Martín-García.

**Data curation:** A. Ignacio Martín-García, Zaira Pardo.

**Formal analysis:** A. Ignacio Martín-García, Zaira Pardo.

**Funding acquisition:** A. Ignacio Martín-García.

**Investigation:** A. Ignacio Martín-García, Alberto Manuel Sánchez-García, Zaira Pardo.

**Methodology:** A. Ignacio Martín-García, Zaira Pardo, Juan Manuel Palma-Hidalgo.

**Project administration:** A. Ignacio Martín-García.

**Resources:** A. Ignacio Martín-García.

**Supervision:** A. Ignacio Martín-García.

**Validation:** A. Ignacio Martín-García, Zaira Pardo, Juan Manuel Palma-Hidalgo.

**Visualization:** A. Ignacio Martín-García, Zaira Pardo.

**Writing – original draft:** A. Ignacio Martín-García, Zaira Pardo.

**Writing – review & editing:** A. Ignacio Martín-García, Alberto Manuel Sánchez-García, Zaira Pardo, Juan Manuel Palma-Hidalgo.

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
