## [Decision Letter · Decision Letter 0]

26 Nov 2024

PONE-D-24-50314Watermelon Plant Silage: A Viable Alternative to Alfalfa Hay for Feeding Murciano-Granadina GoatsPLOS ONE

Dear Dr. Martín-García,

Thank you for submitting your manuscript to PLOS ONE. After careful consideration, we feel that it has merit but does not fully meet PLOS ONE’s publication criteria as it currently stands. Therefore, we invite you to submit a revised version of the manuscript that addresses the points raised during the review process.

We look forward to receiving your revised manuscript.

Kind regards,

Sayed Haidar Abbas Raza

Academic Editor

PLOS ONE

4. Please ensure that you refer to Figure 1 and 2 in your text as, if accepted, production will need this reference to link the reader to the figure.

Additional Editor Comments (if provided):

Reviewers' comments:

Reviewer's Responses to Questions

**Comments to the Author**

1. Is the manuscript technically sound, and do the data support the conclusions?

Reviewer #1: Partly

Reviewer #2: Yes

Reviewer #3: Yes

2. Has the statistical analysis been performed appropriately and rigorously? 

Reviewer #1: Yes

Reviewer #2: Yes

Reviewer #3: Yes

3. Have the authors made all data underlying the findings in their manuscript fully available?

Reviewer #1: Yes

Reviewer #2: Yes

Reviewer #3: No

4. Is the manuscript presented in an intelligible fashion and written in standard English?

Reviewer #1: Yes

Reviewer #2: Yes

Reviewer #3: Yes

5. Review Comments to the Author

Reviewer #1: Considering the in vitro results of this paper, its title is very bold, stating the replacement of alfalfa hay with watermelon silage mainly in terms of energy and dry matter content. It needs to mature and reflect this statement by the authors. Abstract

line 24 replace total volatile fatty acids with short-chain fatty acids throughout the paper. You need to review this conclusion and its relationship with the title of the work.

Introduction:

Line 62 replace palatability with acceptability, a more technical term for animal nutrition

Materials and methods:

In line 74 (Animals and Feeding) what was the concentrate composed of and what was the roughage used in the diet of fistulated animals? There was a mineral mixture, which protein source was used? Diet formulated to meet what nutritional requirements? What variety of watermelon was used? What type of fertilizer did the plants receive (nitrogen, phosphorus and potassium)? This influences the chemical composition of the future silage.

In line 92, wouldn't the objective of formic acid be to reduce the pH and not increase it? Present the pH of watermelon silage after

In line 101 (Experimental Design) you need to make the statistics used clear. A completely randomized design was used with 2 treatments (watermelon silage and alfalfa hay) and 8 incubation times. Was it in a subdivided installment over time? How was this data analyzed?

In line 119 In the second in vitro trial, I ask the same question as in line 101. Detail the experimental design.

The topics (line 101 Experimental Design) and (line 164 Statistical analysis) could be together in the same text to facilitate understanding. How were orthogonal contrasts performed? Detail information.

In line 129, how was crude energy (CE) presented in table 1 determined?

Results:

In line 221, the digestibility of organic matter did not show a effect (p>0.05), so there is no need to say that it was numerically greater, because what matters is the statistical result.

The quality (resolution) of figures 1 and 2 needs to be improved.

Discussion:

In line 334 the authors mention in vitro gas technique, but in the material and methods it is not clear that this technique was used, for example how the pressure was measured at each incubation time.

Conclusions:

The authors conclude that watermelon silage can be used as a suitable ingredient in formulating diets for dairy goats, given the positive effects observed on rumen fermentation and the ability to provide nutrients and energy. In relation to the title of the article “Watermelon Plant Silage: A Viable Alternative to Alfalfa Hay for Feeding Murciano-Granadina Goats” and considering the discussion in line 368 that the second in vitro trial replacing alfalfa hay at 25 and 50% with silage of watermelon did not increase metabolizable energy and the concentration of short-chain fatty acids, I suggest a reformulation in the title of the paper or in this conclusion. It's not coherent.

Reviewer #2: Review: Manuscript (PONE-D-24-50314)

Review of “In vitro fermentation dynamics and microbial change as influenced by different plastic sources combined with urea and molasses using the ruminal fluid of swamp buffalo”

General comments:

In this manuscript the authors aim to propose different silage sources from watermelon plant to replace alfalfa hay in dairy goats diet using in vitro methods. In general, the manuscript addresses an important problem that is often overlooked in animal feeding and environmental sectors. There are numerous strengths to this study, including its diverse sample and well-informed hypotheses.

The topic of the paper well fits within the scope of the journal, but I would consider it for publication only after moderate revisions are performed below:

Abstract:

The abstract seems good and informative; however, it is long and I suggest shortening it.

I would recommend adding an introductory sentence at the start of the abstract.

Keywords:

Keywords should be listed in alphabetical order.

I suggest avoiding the repetitions to increase the chances of the article being found.

Introduction:

Seems good, however, this section should be amended using more recent scientific references.

Material and methods:

The M&M section needs to be supported with updated references.

Results

This section is generally well-written, however, I recommend removing all sentences, as no significant effect was observed.

Discussion

In general, I suggest explaining the results and avoiding comparing them with other studies if it does not impact on the main conclusions.

Conclusion:

Seems good, however, should be re-written avoiding repeating the results and using abbreviations.

References

I urge the authors to update the references.

Tables and figures:

In general, the presentation/communication of tables 1 and 2 requires improvement.

Reviewer #3: This article is an innovative contribution to sustainable livestock farming, exploring the potential use of watermelon plant silage (WPS) as an alternative forage source for dairy goats. The study is well-grounded in the principles of the circular economy, addressing key challenges such as resource scarcity, feed costs, and environmental sustainability. The experimental approach, combining in vitro fermentation trials and comprehensive chemical analyses, provides valuable insights into the nutritional potential and fermentation dynamics of WPS compared to alfalfa hay (AH). The findings suggest that WPS could be a viable alternative feed ingredient, promoting sustainable livestock farming practices while aligning with circular bio-economy principles.

However, before publication, the manuscript requires revisions to address several major and minor issues, as detailed below.

Major Corrections:

1. Strengthen the discussion by clearly linking the findings to their implications for practical livestock feeding strategies, particularly how WPS could mitigate feed shortages or reduce costs in dairy goat production.

Discuss the implications of higher calcium content in WPS and its potential bioavailability, considering its effect on animal health and productivity.

2. Provide more detailed information about the statistical models used, including assumptions and how they were tested. Clarify why Fisher’s LSD test was chosen and consider additional tests if necessary.

Justify the significance thresholds used (P < 0.05 for significance and 0.10 for trends) and ensure consistency throughout the text.

3. Highlight the limitations of in vitro experiments and the need for in vivo validation to confirm the practical applicability of WPS in dairy goat diets. Expand on how these findings could translate to field conditions.

Environmental Impact Assessment:

4. Include a discussion on the environmental implications of using WPS, such as reduced waste and potential reductions in greenhouse gas emissions compared to conventional forages.

5. Improve the clarity of figures, such as cumulative gas production graphs, ensuring all labels, legends, and axes are legible.

Minor Corrections:

1. Correct grammatical errors, such as “...the increasing of the prices…” (line 47) to “...the increase in prices.”

Ensure consistency in terminology (e.g., use either "watermelon plant silage" or "WPS" uniformly throughout).

2. Specify the methodology used for measuring pH, VFAs, and other parameters to ensure replicability.

Clarify whether the AH used in the study was sourced from a single batch or multiple batches.

3. Ensure consistent formatting of tables, such as proper alignment and column headers.

Standardize the citation format for references, particularly in-text citations.

4. Expand the introduction to include more references to studies on alternative silage materials, providing a broader context for the research.

5. Include a more direct call to action for in vivo trials and practical implementation studies to further explore the benefits of WPS in dairy goat diets.

Addressing these revisions will significantly enhance the clarity, impact, and practical relevance of the manuscript, making it suitable for publication.

6. PLOS authors have the option to publish the peer review history of their article (what does this mean? ). If published, this will include your full peer review and any attached files.

**Do you want your identity to be public for this peer review?** For information about this choice, including consent withdrawal, please see our Privacy Policy .

Reviewer #1: No

Reviewer #2: No

Reviewer #3: **Yes: ** Simna Saraswathi Prasannakumari

---

## [Author Response · Author response to Decision Letter 1]

25 Mar 2025

Reviewer #1: Considering the in vitro results of this paper, its title is very bold, stating the replacement of alfalfa hay with watermelon silage mainly in terms of energy and dry matter content. It needs to mature and reflect this statement by the authors.

• Authors: The title has been rewritten following the indications of R#1: “Chemical composition and in vitro nutritional assessment of watermelon (Citrullus lanatus) plant silage as a forage option for Murciano-Granadina goats”.

Abstract

line 24 replace total volatile fatty acids with short-chain fatty acids throughout the paper. You need to review this conclusion and its relationship with the title of the work.

• Authors: The term “total volatile fatty acids” has been replaced with “short-chain fatty acids” throughout the paper. Consequently, the abbreviation VFA has been replaced by SCFA.

Introduction:

Line 62 replace palatability with acceptability, a more technical term for animal nutrition

• Authors: The term “palatability” has been replaced with “acceptability”, the only time this word is mentioned throughout the text.

Materials and methods:

In line 74 (Animals and Feeding) what was the concentrate composed of and what was the roughage used in the diet of fistulated animals? There was a mineral mixture, which protein source was used?

• Authors: Clarifications regarding the doubts about the feeding of animals donating ruminal inoculum, rightly raised by R#1, have been included in the text, in the Animals and Feeding section

Diet formulated to meet what nutritional requirements?

• Authors: The paper justifying the ration employed to cover the “protein and energy” requirement of this breed has now been cited in the Animals and Feeding section, and included in the references list of the manuscript. (Prieto et al. 1990. Protein and energy requirements for maintenance of indigenous Granadina goats, Brit. J. Nutr. 63 155–163).

What variety of watermelon was used?

• Authors: “Sugar Baby” variety of watermelon was employed. This information has now been included in the beginning of the Silage Preparation section.

What type of fertilizer did the plants receive (nitrogen, phosphorus and potassium)? This influences the chemical composition of the future silage.

• Authors: “The watermelon growing area was fertilized with 175 kg/ha of an NPK mixture (ratio 2.6:0.8:3.4) before planting.” This sentence has now been included in the beginning of the Silage Preparation section of the manuscript.

In line 92, wouldn't the objective of formic acid be to reduce the pH and not increase it? Present the pH of watermelon silage after

• Authors: Yes, R#1 is right. Formic acid was used to enhance the pH reduction. The sentence has been corrected to make it easier to read (included in the Silage Preparation section): “Formic acid (0.45% of fresh matter) was previously added to facilitate the drop in pH”.

In line 101 (Experimental Design) you need to make the statistics used clear. A completely randomized design was used with 2 treatments (watermelon silage and alfalfa hay) and 8 incubation times. Was it in a subdivided installment over time? How was this data analyzed?

• Authors: Following the indications of the R#1, the design of Experiment 1 has been clarified in the text (second paragraph in the Experimental Design and Samplings section): “A complete randomized design was used in the first trial, and the main effect was the type of forage (AH and WPS). The same amount of DM (300 mg) of each forage was carefully weighed into 120 ml Wheaton bottles. Twelve bottles were prepared per type of forage (four sources of inoculum in triplicate)”, and: “These data were used to calculate the kinetics of gas production from forage fermentation as described later.”

Data resulted from this trial were analysed as described in “Statistical analysis and calculations” section: “The comparison between AH and WPS fermentation parameters was analysed by a one-way analysis of variance (ANOVA)”

In line 119 In the second in vitro trial, I ask the same question as in line 101. Detail the experimental design.

The topics (line 101 Experimental Design) and (line 164 Statistical analysis) could be together in the same text to facilitate understanding. How were orthogonal contrasts performed? Detail information.

• Authors: Following the indications of the R#1, the design of Experiment 2 has been clarified in the text (Line 130 and forward): “The second in vitro trial evaluated the effect of replacing AH with WPS in different proportions using a completely randomized design, and the main effect was the substitution rate: 0, 25 and 50%.)”, and Lines 134-136: “In vitro incubations were prepared following the same protocol as in the first trial, with 12 bottles per level of AH substitution by WPS (four inoculum sources in triplicate).”

Data resulted from this second trial were analysed as described in “Statistical analysis and calculations” section (Lines 182-): “The effect of substituting AH with WPS was also analysed with ANOVA, followed by orthogonal polynomial contrast to evaluate linear trends in the effect of increasing substitution levels”. The results of this analysis have been described in the Results and Discussion sections.

In line 129, how was crude energy (CE) presented in table 1 determined?

• Authors: It was certainly a mistake on our part. The description of the method for determining Crude Energy (CE) is now detailed in the text on the final part of the first paragraph of the Chemical Analyses section: “The gross energy (GE) content was measured determined by using an adiabatic calorimeter (Parr Instruments Co. model 1356, Moline, IL, USA).”

Additionally, the text has been improved by replacing the term “crude energy” with “gross energy”, which is more appropriate for animal nutrition studies.

Results:

In line 221, the digestibility of organic matter did not show a effect (p>0.05), so there is no need to say that it was numerically greater, because what matters is the statistical result.

• Authors: Thank you for your valuable observation. Text has been corrected and now is described as: “In contrast, OMD was similar for WPS and AH. (471 and 410 g/kg, respectively; P = 0.126).”

The quality (resolution) of figures 1 and 2 needs to be improved.

• Authors: I apologize. It was really a mistake. Figures 1 and 2 have been improved in quality and are attached as new files. Additionally, we have checked their resolution in the PACE platform as indicated in the journal

Discussion:

In line 334 the authors mention in vitro gas technique, but in the material and methods it is not clear that this technique was used, for example how the pressure was measured at each incubation time.

• Authors: Thank you for your valuable comment. If I have misunderstood your statement, I sincerely apologize. However, we believe that the procedure for measuring pressure and volume of gas production has been described in the 'Experimental Design and Sampling' section (second paragraph). Nonetheless, we have made some revisions to enhance the clarity of the text.

Conclusions:

The authors conclude that watermelon silage can be used as a suitable ingredient in formulating diets for dairy goats, given the positive effects observed on rumen fermentation and the ability to provide nutrients and energy. In relation to the title of the article “Watermelon Plant Silage: A Viable Alternative to Alfalfa Hay for Feeding Murciano-Granadina Goats” and considering the discussion in line 368 that the second in vitro trial replacing alfalfa hay at 25 and 50% with silage of watermelon did not increase metabolizable energy and the concentration of short-chain fatty acids, I suggest a reformulation in the title of the paper or in this conclusion. It's not coherent.

• Authors: I appreciate your precise correction. In accordance to the revised tittle, conclusion has been modified following your suggestion:

“This study shows that the nutritional value of watermelon plant silage suggests it may be suitable for use as a forage component in the formulation of diets for dairy goats. This claim would be based both on its chemical composition and on the fact that, compared to alfalfa hay as a reference forage, its inclusion in the ration did not alter the different parameters of in vitro ruminal fermentation and the supply of nutrients and energy. This evidence supports the use of watermelon plant silage as an alternative source in ruminant feeding, although in vivo studies are required to elucidate the influence of this fodder on voluntary feed intake and animal performance and production.

In addition, the inclusion of watermelon plant silage would contribute to the circular bioeconomy by utilizing agricultural waste as a feed source for livestock. This approach would help alleviate the problem of shortage of conventional fodder while potentially reducing costs for livestock farmers.”

Reviewer #2: Review: Manuscript (PONE-D-24-50314)

Review of “In vitro fermentation dynamics and microbial change as influenced by different plastic sources combined with urea and molasses using the ruminal fluid of swamp buffalo”

General comments:

In this manuscript the authors aim to propose different silage sources from watermelon plant to replace alfalfa hay in dairy goats diet using in vitro methods. In general, the manuscript addresses an important problem that is often overlooked in animal feeding and environmental sectors. There are numerous strengths to this study, including its diverse sample and well-informed hypotheses.

The topic of the paper well fits within the scope of the journal, but I would consider it for publication only after moderate revisions are performed below:

Abstract:

The abstract seems good and informative; however, it is long and I suggest shortening it.

I would recommend adding an introductory sentence at the start of the abstract.

• Authors: I'm grateful for your constructive remarks. Abstract has been considerably shorted (from 365 to 286 words) and now contains an introductory statement.

Keywords:

Keywords should be listed in alphabetical order. I suggest avoiding the repetitions to increase the chances of the article being found

• Authors: A Keyword section has been listed (by-products nutritive value evaluation, circular economy, forages, in vitro rumen fermentation, watermelon plant silage).

Introduction:

Seems good, however, this section should be amended using more recent scientific references.

• Authors: The Introduction section has been modified to contain updated references where possible.

Material and methods:

The M&M section needs to be supported with updated references.

• Authors: As a result of the application of the observations of the different reviewers, the Material and Methods section has been substantially modified and, when possible, updated references have been referred.

Results

This section is generally well-written, however, I recommend removing all sentences, as no significant effect was observed.

• Authors: Thank you for your feedback. We appreciate your comments and, to address them, in the Results section, all comments mentioning "numerically” differences have been removed even though no statistically significant differences (P>0.05) or trends to be so (P>0.1) were found. Nevertheless, let us propose that since this is a nutritional evaluation study of WPS versus another reference forage, such as alfalfa hay, the authors consider it is important for the reader to highlight the lack of significant differences in key parameters of the nutritional value. This would also apply when mentioning the lack of differences when forage replaces conventional forage (experiment 2). In such cases, the absence of differences would be "good news" for the hypothesis that WPS can be an alternative forage.

We believe the proposed modifications comprehensively respond to the reviewer's observations. We remain open to further discussion and refinement of our manuscript.

Discussion

In general, I suggest explaining the results and avoiding comparing them with other studies if it does not impact on the main conclusions.

• Authors: Thank you very much particularly for this comment, which clearly indicates your successful commitment to clarity and coherence between the initial hypothesis, objectives, and conclusions of the scientific work. Nonetheless, following your advices and those of the other reviewers, the title, objectives, and conclusions have been modified, and consequently, the discussion has been rewritten, which I believe represents, in general, a clear improvement of the article. I hope that the discussion is now more closely aligned with the suggestion you made in this aspect.

Conclusion:

Seems good, however, should be re-written avoiding repeating the results and using abbreviations.

• Authors: In order to collect your instructions along with those of another reviewer, the Conclusions section has been rewritten.

References

I urge the authors to update the references.

• Authors: Several of references have been updated, especially those used in the Introduction section.

Tables and figures:

In general, the presentation/communication of tables 1 and 2 requires improvement.

• Authors: Tables 1 and 2 have been modified following the reviewers’ recommendations.

Reviewer #3: This article is an innovative contribution to sustainable livestock farming, exploring the potential use of watermelon plant silage (WPS) as an alternative forage source for dairy goats. The study is well-grounded in the principles of the circular economy, addressing key challenges such as resource scarcity, feed costs, and environmental sustainability. The experimental approach, combining in vitro fermentation trials and comprehensive chemical analyses, provides valuable insights into the nutritional potential and fermentation dynamics of WPS compared to alfalfa hay (AH). The findings suggest that WPS could be a viable alternative feed ingredient, promoting sustainable livestock farming practices while aligning with circular bio-economy principles.

However, before publication, the manuscript requires revisions to address several major and minor issues, as detailed below.

Major Corrections:

1. Strengthen the discussion by clearly linking the findings to their implications for practical livestock feeding strategies, particularly how WPS could mitigate feed shortages or reduce costs in dairy goat production.

• Authors: I appreciate your recommendation to cover this important feature. A paragraph discussing this aspect has been inserted near the end of the Discussion section.

Discuss the implications of higher calcium content in WPS and its potential bioavailability, considering its effect on animal health and productivity.

• Authors: Thank you for your comment. It really improves the article a lot by adding your suggestions. Following the discussion of the nature of WPS calcium, we have added additional text discussing the aspects you mention.

2. Provide more detailed information about the statistical models used, including assumptions and how they were tested. Clarify why Fisher’s LSD test was chosen and consider additional tests if necessary.

• Auth.: We appreciate the reviewer’s comment. We have expanded the "Statistical Analysis" section: “Data analysis was performed using SPSS software (IBM Corp. IBM SPSS Statistics for Windows, Version 29.0.0.0, Armonk, New York, USA). The comparison between AH and WPS fermentation parameters was analyzed by one-way ANOVA. Normality and homoscedasticity assumptions for ANOVA were verified using the Shapiro-Wilk test (p > 0.10 for all variables) and Levene’s test (p > 0.05), respectively. The effect of substituting AH with WPS was assessed via ANOVA with orthogonal polynomial contrasts to evaluate linear trends across increasing substitution levels. These analyses examined the impact of forage type and substitution levels on degradation parameters (GP, ME, OMD, pH, SCFA, and CH₄ production). Fisher’s Least Significant Difference (LSD) test was selected for planned pairwise comparisons following significant ANOVA results (p < 0.05). No unplanned comparisons were conducted, maintaining the per-comparison error rate (α = 0.05). Trends were noted for P values between 0.0

---

## [Decision Letter · Decision Letter 1]

10 Apr 2025

Chemical composition and in vitro nutritional assessment of watermelon (Citrullus lanatus) plant silage as a forage option for Murciano-Granadina goats

PONE-D-24-50314R1

Dear Dr. Martín-García,

We’re pleased to inform you that your manuscript has been judged scientifically suitable for publication and will be formally accepted for publication once it meets all outstanding technical requirements.

Kind regards,

Sayed Haidar Abbas Raza

Academic Editor

PLOS ONE

Additional Editor Comments (optional):

Reviewers' comments:

Reviewer's Responses to Questions

**Comments to the Author**

1. If the authors have adequately addressed your comments raised in a previous round of review and you feel that this manuscript is now acceptable for publication, you may indicate that here to bypass the “Comments to the Author” section, enter your conflict of interest statement in the “Confidential to Editor” section, and submit your "Accept" recommendation.

Reviewer #2: All comments have been addressed

Reviewer #3: All comments have been addressed

2. Is the manuscript technically sound, and do the data support the conclusions?

Reviewer #2: Yes

Reviewer #3: Yes

3. Has the statistical analysis been performed appropriately and rigorously? 

Reviewer #2: Yes

Reviewer #3: Yes

4. Have the authors made all data underlying the findings in their manuscript fully available?

Reviewer #2: Yes

Reviewer #3: (No Response)

5. Is the manuscript presented in an intelligible fashion and written in standard English?

Reviewer #2: Yes

Reviewer #3: Yes

6. Review Comments to the Author

Reviewer #2: The authors have addressed my comments raised in the previous review, however English editing is still needed.

Reviewer #3: (No Response)

7. PLOS authors have the option to publish the peer review history of their article (what does this mean? ). If published, this will include your full peer review and any attached files.

**Do you want your identity to be public for this peer review?** For information about this choice, including consent withdrawal, please see our Privacy Policy .

Reviewer #2: No

Reviewer #3: No

---

## [Editor Report · Acceptance letter]

PONE-D-24-50314R1

PLOS ONE

Dear Dr. Martín-García,

I'm pleased to inform you that your manuscript has been deemed suitable for publication in PLOS ONE. Congratulations! Your manuscript is now being handed over to our production team.

Kind regards,

on behalf of

Dr. Sayed Haidar Abbas Raza

Academic Editor

PLOS ONE